# Chest CT Findings and SARS-CoV-2 Infection in Trauma Patients—Is There a Prediction towards Higher Complication Rates?

**DOI:** 10.3390/jcm11216401

**Published:** 2022-10-29

**Authors:** Fabrice Scheurer, Sascha Halvachizadeh, Till Berk, Hans-Christoph Pape, Roman Pfeifer

**Affiliations:** Division of Trauma Surgery, Department of Surgery, University Hospital Zurich, University of Zurich, 8091 Zurich, Switzerland

**Keywords:** polytrauma, treatment strategy, CT thorax, SARS-CoV-2, trauma, outcome, pulmonary complications

## Abstract

Background: Polytrauma patients with SARS-CoV-2 infections may be associated with an increased complication rate. The main goal of this study was to analyze the clinical course of trauma patients with COVID infection and a positive CT finding. Methods: This was a retrospective in-hospital study. Polytrauma patients diagnosed with SARS-CoV-2 infections were included in our analysis. The outcome parameters were pulmonary complication during admission, pulmonary embolism, pleural effusion, pneumonia, mortality, length of stay and readmission < 30 days. Results: 48 patients were included in the study. Trauma patients in the age-adjusted matched-pair analysis with typical changes in SARS-CoV-2 infection in CT findings showed significantly more pulmonary complications in general and significantly more cases of pneumonia (complications: 56% vs. 11%, *p* = 0.046; pneumonia 44% vs. 0%, *p* = 0.023). In addition, the clinical course of polytrauma patients with SARS-CoV-2 infection showed a high rate of pulmonary complications in the inpatient course (53%). Conclusion: The results of our study show that the changes in the CT findings of trauma patients with SARS-CoV-2 infection are a good indicator of further inpatient outcomes. Similarly, polytrauma patients with a SARS-CoV-2 infection and positive CT findings are shown to have increased risk for pulmonary complications.

## 1. Introduction

Severe acute respiratory syndrome coronavirus 2 (SARS-CoV-2) has become one of the world’s most worldwide medical problems due to its global reach and the fact that more than 500 million people have already been infected. More than 5 million people have died since the start of the SARS-CoV-2 pandemic [1]. Furthermore, the current mortality rate of 2.2% continues to be very serious given the large number of SARS-CoV-2 cases [2]. Furthermore, patients hospitalized with SARS-CoV-2 show an increased incidence of complications (cerebrovascular disorders, encephalopathy, acute respiratory distress syndrome (ARDS), pneumonia, myocarditis, arrhythmias, acute liver failure, vasculitis, and coagulopathies) [2].

There are still significantly less data and guidelines regarding surgeries and emergency interventions in trauma patients with SARS-CoV-2 compared with other diseases [3,4]. Thus, it has already been shown that postoperative mortality increases when patients have already been infected with SARS-CoV-2 at the time of surgery [5]. Especially for polytrauma patients with SARS-CoV-2 infection, there are still very few data. Polytrauma patients are known to be very susceptible to complications, such as shock, acidosis, and coagulopathy during the inpatient course [6]. It has been shown that trauma is one of the most common 10 causes of death worldwide [7]. However, treatment strategies in polytrauma patients infected with SARS-CoV-2 have not been discussed so far. It is not known whether this group of patients benefit from a damage control procedure [8,9,10] in case of infection with SARS-CoV-2. Therefore, the primary aim of this study is to find out whether polytrauma patients with an additional SARS-CoV-2 infection develop more complications during the inpatient course. For this aim, we analyzed the clinical course of polytrauma patients with a SARS-CoV-2 infection. Additionally, our goal was to find out if there was a possible correlation of SARS-CoV-2 typical changes in CT findings on admission and the following complications during the inpatient course of trauma patients.

## 2. Materials and Methods

### 2.1. Study Subjects and Study Design

This study is a retrospective data analysis and was approved by the local ethics committee (KEK-ZH-Nr. 2020-01037). The data for the outcome parameters were extracted from the medical records of patients in the electronic hospital information system. The data were collected over a total period of 2 years, from March 2019 to March 2021. Included were the beta, gamma, and delta variants of SARS-CoV-2. The inclusion criteria were that all patients had a positive SARS-CoV-2 test on the day of admission and trauma as their principal diagnosis at the time of admission. The test method was in every case a polymerase chain reaction (PCR) nasopharyngeal swab. All of the patients were admitted via the trauma bay or the emergency department. All patients were older than 18 years at the time of admission.

Exclusion criteria were patients with a previous history of lung surgery, patients with bacterial lung diseases, patients with lung carcinomas or patients with lung carcinomas in remission, as well as when patients were referred or repatriated from other hospitals or if there was the presence of a documented refusal. Patients were divided into two groups according to the Injury Severity Score (ISS) [11,12]. One group included all patients infected with SARS-CoV-2 who had an ISS ≥ 16. The other group included all trauma patients with an ISS < 16 and a SARS-CoV-2 infection.

For the secondary study question, only the patients who were admitted via the trauma bay and consequently received a whole-body computed tomography (CT) scan were included. Again, the patients were divided into two groups: the CT (+) group and CT (−) group. In the CT (+) group, all trauma patients who were infected with SARS-CoV-2 and who had SARS-CoV-2-related changes in CT findings were included. In the CT (−) group, all trauma patients who were infected with SARS-CoV-2 but without SARS-CoV-2-related changes in CT findings were included.

The outcome parameters were total of pulmonary complications during admission, pulmonary embolism, pleural effusion, pneumonia, mortality during admission, length of stay, body mass index (BMI) > 25 kg/m^2^ [13] and readmission < 30 days. Other data that were collected included demographics (age and gender), ISS [11], and Abbreviated Injury Scale (AIS) [14] (head/neck, chest, abdomen, pelvis/extremity, and external).

### 2.2. Definitions

A polytrauma patient is defined by an ISS ≥ 16 [11]. The ISS and AIS for this study were calculated using the definitive diagnoses in the discharge reports. For the specific complication of pneumonia, we defined that patients must have at least two of the three main symptoms to be considered positive: fever > 38.5 °C, dyspnea or tachypnea and cough. The SARS-CoV-2 typical changes in CT findings were extracted from the CT findings reports in the electronic hospital information system and were classified according to the Radiological Society of North America expert consensus statement on reporting chest CT findings related to COVID-19 [15]. The CT (−) group included “negative for pneumonia”, and the CT (+) group included “atypical appearance,” “indeterminate appearance”, and “typical appearance.” The age difference was too big; therefore, an additional age-adjusted matched-pair analysis was performed to correct this bias. For this purpose, we compared the nine youngest people from the CT (+) group with the 9 oldest people from the CT (−) group.

### 2.3. Statistics

It is a descriptive statistic with a retrospective data analysis. Continuous parameters were compared using Student’s *t*-test. Discrete variables were compared using Pearson’s chi-square test. A significance level of *p* < 0.05 applies to both tests. For the calculation, the IBM SPSS program was used (version 26, licensors 1989, 2019).

## 3. Results

### Patient Data and Demographics

A total of 48 patients with trauma as the principal diagnosis and a positive SARS-CoV-2 PCR result on admission were included in the study. There were 33 male patients (69%) and 15 female patients (31%), with a mean age of 60.3 years (SD 19.3 years). Of these patients, 15 of them (31%) were polytrauma patients with an ISS > 16. The median ISS was 10 (SD 7.0). Of all the patients, 34 of them (71%) received a whole-body CT on admission. 

We analyzed the pulmonary complications of polytrauma patients with SARS-CoV-2 infection on admission during their inpatient course and identified a high number of pulmonary complications (53%). There were especially high rates of pneumonia (53%) and pulmonary embolism (20%) among inpatients, as well 33% of all polytrauma patients developed a pleural effusion during their inpatient course (Table 1).

In all patients (n = 34) who received a whole-body CT scan we analyzed whether SARS-CoV-2 typical changes in CT findings were already present at admission (Figure 1). In the CT (+) group, there were significantly more pulmonary complications (*p* = 0.001), more pneumonias (*p* = 0.002) and more cases of pleural effusion (*p* = 0.027) than in the CT (-) group (Table 2).

There was no significant difference in pulmonary embolism (*p* = 0.141), despite changes typical of SARS-CoV-2 infection in CT findings being present. There were also no significant differences regarding length of stay (*p* = 0.171), mortality during admission (*p* = 0.210), readmission < 30 days (*p* = 0.952) and BMI > 25 kg/m^2^ (*p* = 0.215).

Due to the difference in mean age among these groups (69.3 y/37.1 y, *p* = 0.001), an age-adjusted matched-pair analysis was performed. The age-adjusted matched-pair analysis (55.3 y/46.5 y, *p* = 0.165) showed significantly more total pulmonary complications (*p* = 0.046) and pneumonias (*p* = 0.023) in the CT (+) group with typical changes in SARS-CoV-2 in CT findings. There was no longer a significant difference in pulmonary embolism (*p* = 0.580) and pleural effusion (*p* = 0.114). There were no significant differences regarding pulmonary embolism (*p* = 0.580), pleural effusion (*p* = 0.114), length of stay (*p* = 0.165), mortality during admission (*p* = 1.0), readmission < 30 days (*p* = 1.0) and BMI > 25 kg/m^2^ (*p* = 0.637) (Table 3).

## 4. Discussion

There are already many studies on guidelines for the treatment and management of severely injured patients in the inpatient setting [12,16,17,18]. In addition, emergency treatment in the trauma bay is standardized according to the Advanced Trauma Life Support (ATLS) guidelines [19]. In a short period, several studies have been published for treatment strategies in patients with a SARS-CoV-2 infection [20,21]. However, treatment concepts for polytrauma patients with a SARS-CoV-2 infection do not exist. 

This study has shown the following main results:

Polytrauma patients with SARS-CoV-2 have shown a very high complication rate (pneumonia 53%, lung embolism 20%). This is significantly higher compared with isolated polytrauma patients with blunt chest trauma without infection with a pneumonia rate of 12.7% [22] and lung embolism rate of 9–12% [23].

It has also been shown that trauma patients with a SARS-CoV-2 infection at admission and typical changes in SARS-CoV-2 in CT findings at the primary CT scan of the chest have significantly more total pulmonary complications during admission (*p* = 0.046) and pneumonias (*p* = 0.023) than the comparison group that had an initial CT scan of the chest that was negative for typical changes in SARS-CoV-2.

For the first study question, it can be concluded that it is essential to assess the risk of pulmonary embolism in polytrauma patients at a minimum from the beginning. It has already been shown in the literature that mortality is significantly higher in patients who have a SARS-CoV-2 infection in addition to a pulmonary embolism [24]. It is also already known that polytrauma patients suffer more complications during hospitalization [10,25] than monotrauma patients. Large-scale multicenter retrospective data analysis was performed for 2746 patients in whom pulmonary emboli were studied after complex trauma in a time period over 30 months. In this study, complex trauma was defined in such a way that all patients had to be transferred to the intensive care unit after major trauma. In this patient population, 4.6% of all patients had a pulmonary embolism during their inpatient course [26,27]. Although this is a very similar patient population in our study with a SARS-CoV-2 infection, there were clearly fewer cases of pulmonary embolism than in our polytrauma patients with an additional SARS-CoV-2 infection (20%). In polytrauma patients with a SARS-CoV-2 infection, there are several options to counteract pulmonary embolism in terms of primary prophylaxis. 

To develop preventive measures, the pathophysiology of coronavirus thrombogenesis must first be looked at. To infect a host, the virus uses the angiotensin-converting enzyme receptors on the surface of pneumocytes in the epithelial alveolar surface [28]. Due to the inflammatory reaction, an integrity defect with increased permeability and leakage occurs in the alveoli [29]. Thus, the picture of disseminated intravascular coagulation is formed [30]. Knowing this, it essential to ensure that prophylactic anticoagulation in polytrauma patients with SARS-CoV-2 infection is administered as early as possible and as soon as the injury pattern allows it, which must be looked at and decided individually in each case. The known risk factors for pulmonary embolism are high, but can be summarized into the following three groups: genetic, acquired and environmental risk factors [31]. Despite the genetic and acquired risk factors not being able to be influenced, we have the opportunity to intervene in the environmental risk factors as described in the text above. Furthermore, care should be taken to ensure that patients are mobilized as early as possible during their inpatient course to avoid a long immobilization phase. This can be achieved by consistently trying to achieve definitive surgical treatment as early as possible, if the patient’s health situation allows it. Another factor to keep in mind is that the tendency to swell allows for a primary definitive surgical procedure. 

In addition, care should be taken to include physical therapy in the treatment plan as early as possible in this patient population—on the one hand, to ensure early functional mobilization, which corresponds to thromboembolism prophylaxis, and on the other hand, that all these patients receive respiratory physiotherapy to prevent pneumonia.

Knowing from our second study question that the trauma patients with CT findings are already at higher risk for pulmonary complications, the aim could be to avoid a second hit if possible or to keep it at least as small as possible. In concrete terms, damage control orthopedic surgery, for example, an external fixator, might be advantageous in this patient collective. However, there is a gray balance between minimizing the risk of pulmonary complications and the longer immobilization phase with delayed surgical treatment.

For the daily hospital routine, this also means that, in this patient population, the required diagnostics should be performed in cases of clinical suspicion for a lung embolism as soon as possible [24,32]. It will also be important to detect the clinical and laboratory signs of pneumonia early in order to prevent it or to be able to initiate appropriate therapy [33,34,35]. For the following outcome parameters: mortality during admission, length of stay, obesity, and readmission < 30 days, no significant difference was found in our results related to pulmonary complications in the inpatient setting. Thus, no preventive measures can be derived from these parameters. This is most likely due to the fact that the long-term effects of SARS-CoV-2 infection were not considered in this study.

## 5. Conclusions

The results of this study have shown that polytrauma patients who have a concomitant SARS-CoV-2 infection and positive CT findings are highly susceptible to pulmonary complications during their inpatient course. Therefore, a major focus should be to aim for early mobilization and sufficient anticoagulation to achieve prophylaxis for pneumonia and pulmonary embolism.

### Limitations

This was a single-center retrospective study with a small number of patients. It was difficult for a single hospital to obtain a large number of trauma patients who also tested positive for SARS-CoV-2 on the day of admission. In the future, a multicenter study would certainly be useful to obtain a larger number of patients. Additionally, the Omicron variant was not included in our study. Here, a comparison would certainly be interesting to detect potential differences.

## Figures and Tables

**Figure 1 jcm-11-06401-f001:**
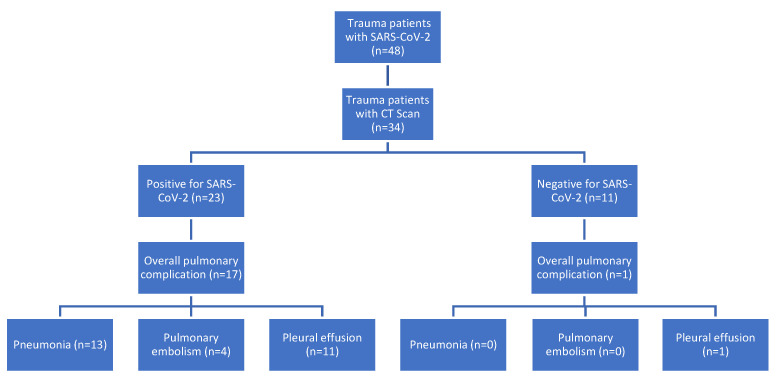
Clinical course for trauma patients with SARS-CoV-2 infection and positive CT findings.

**Table 1 jcm-11-06401-t001:** Trauma patients with SARS-CoV-2 infection.

		ISS ≥ 16	ISS < 16
Count, n	48	15	33
Gender, n = male (%)	33 (69)	13 (87)	20 (61)
Mean age, (SD)	60.3 (19.3)	55.0 (18.4)	62.7 (19.6)
Median ISS, (SD)	10 (7.0)	22.0 (4.7)	9.0 (2.6)
Median AIS head/neck, (SD)	2 (1.34)	2.5 (1.1)	1.3 (1.3)
Median AIS chest, (SD)	0 (1.0)	0.8 (1.37)	0.2 (0.73)
Median AIS abdomen, (SD)	0 (1.8)	1.5 (2.0)	0.2 (1.6)
Median AIS pelvis/extremity, (SD)	0 (1.9)	1.5 (1.9)	1.3 (2.0)
Median AIS external, (SD)	0 (1.92)	0.7 (0.7)	0.4 (0.6)
Whole-body CT on admission (%)	34 (71)	15 (100)	19 (56)
Overall pulmonary complications, n (%)	19 (40)	8 (53)	11 (34)
Pulmonary embolism, n (%)	4 (8)	3 (20)	1 (3)
Pneumonia, n (%)	14 (29)	8 (53)	6 (19)
Pleural effusion, n (%)	13 (27)	5 (33)	8 (25)
mortality during admission, n (%)	4 (8)	1 (7)	3 (9)
Readmission < 30 days, n (%)	3 (6)	1 (7)	2 (6)
Length of stay, mean days (SD)	9.6 (8.4)	11 (7.9)	9.4 (8.7)
BMI > 25 kg/m^2^, n (%)	18 (38)	7 (47)	11 (34)

**Table 2 jcm-11-06401-t002:** Clinical course for polytrauma patients with SARS-CoV-2 infection and positive CT findings.

		CT Findings Positive for SARS-CoV-2 Pneumonia	CT Findings Negative for SARS-CoV-2 Pneumonia	*p* Value
Count (Whole-body CT on admission), n	34	23	11	
Median ISS, (SD)	9 (7.2)	14.1 (6.9)	17.5 (7.5)	0.213
Gender, n = male (%)	28 (82)	19 (83)	9 (82)	
Mean age, (SD)	59.5 (19.1)	69.3 (14.5)	37.1 (14.8)	0.001
Median AIS head/neck, (SD)	2 (1.2)	2.2 (1.3)	1.8 (1.3)	0.567
Median AIS chest, (SD)	0 (1.2)	0.5 (1.2)	0.3 (1.2)	0.957
Median AIS abdomen, (SD)	0 (2.1)	0.8 (2.2)	0.9 (2.0)	0.822
Median AIS pelvis/extremity, (SD)	0 (2.1)	0.8 (1.4)	1.6 (2.9)	0.077
Median AIS external, (SD)	0 (0.7)	0.4 (0.6)	0.7 (0.8)	0.137
Overall pulmonary complications, n (%)	18 (53)	17 (74)	1 (9)	0.001
Pulmonary embolism, n (%)	4 (12)	4 (17)	0 (0)	0.141
Pneumonia, n (%)	13 (38)	13 (57)	0 (0)	0.002
Pleural effusion, n (%)	12 (35)	11 (48)	1 (9)	0.027
Mortality during admission, n (%)	3 (9)	3 (13)	0 (0)	0.210
Readmission < 30 days, n (%)	3 (9)	2 (9)	1 (9)	0.952
Length of stay, mean days (SD)	10.6 (9.0)	13.1 (10.4)	7.5 (3.5)	0.171
BMI > 25 kg/m^2^, n (%)	13 (38)	9 (39)	4 (34)	0.215

**Table 3 jcm-11-06401-t003:** Age-adjusted pair-matched analysis of polytrauma patients with SARS-CoV-2 infection and positive CT findings.

		CT Findings Positive for SARS-CoV-2 Pneumonia	CT Findings Negative for SARS-CoV-2 Pneumonia	*p* Value
Count, n	18	9	9	
Median ISS, (SD)	16.1 (7.7)	16.0 (7.2)	18.2 (8.0)	0.256
Gender, n = male (%)	17	9 (100)	8 (89)	
Mean age, (SD)	49.8 (9.9)	55.3 (7.5)	46.5 (11.2)	0.165
Median AIS head/neck, (SD)	1.7 (1.2)	1.6 (1.3)	1.6 (1.2)	0.854
Median AIS chest, (SD)	0.5 (1.2)	0.7 (1.3)	0.3 (1.0)	0.556
Median AIS abdomen, (SD)	1.6 (2.6)	1.4 (3.1)	1.1 (2.2)	0.438
Median AIS pelvis/extremity, (SD)	1.7 (2.5)	0.7 (1.3)	2.0 (3.0)	0.089
Media AIS external, (SD)	0.6 (0.7)	0.4 (0.5)	0.7 (0.8)	0.326
Overall pulmonary complications, n (%)	6 (33)	5 (56)	1 (11)	0.046
Pulmonary embolism, n (%)	2 (11)	2 (22)	0 (0)	0.580
Pneumonia, n (%)	4 (22)	4 (44)	0 (0)	0.023
Pleural effusion, n (%)	5 (28)	4 (44)	1 (11)	0.114
Mortality during admission, n (%)	0 (0)	0 (0)	0 (0)	1.000
Readmission < 30 days, n (%)	2 (11)	1 (11)	1 (11)	1.000
Length of stay, mean days (SD)	11.9 (9.3)	15.0 (12.6)	8.8 (2.4)	0.165
BMI > 25 kg/m^2^, n (%)	9 (50)	5 (56)	4 (44)	0.637

## Data Availability

The datasets generated during and/or analyzed during the current study are available from the corresponding author on reasonable request.

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
