# Peer review of "Chest CT Findings and SARS-CoV-2 Infection in Trauma Patients—Is There a Prediction towards Higher Complication Rates?"

_jcm, 2022, doi:10.3390/jcm11216401_

Round 1

Reviewer 1 Report

Dear editors, thank you for asking me to review this study. it is quite a simple study, but one that adds to the literature around these patients. it is not out of keeping with larger data sets in the literature, for example the covid surg datasets.  the number of patients is small, but the analysis has been done well. simple descriptive stats are appropriate, and the authors take care not to infer many results.

minor point

1. it might be worth stating the type of covid this was, was this delta or omicron? as they are 2 totally different entities, with the first causing much more respriatory failure, but the latter being found more incidentally. this will make the study more generalisable. 

thanks again

Author Response

Dear Sir or Madam

Thank you very much for reviewing my article. Included were the beta, gamma, and delta variants of SARS-CoV-2. I included it in the article

kind regars

Fabrice Scheurer

Reviewer 2 Report

Thank you for reviewing the manuscript of F. Scheurer et al. entitled “Chest CT findings and SARS-CoV-2 infection in trauma patients – Is there a prediction towards higher complication rates?”. To summarize, Scheurer and colleges prepared a retrospective study in SARS-CoV-2 positive (poly)trauma patients and showed a higher risk of pulmonary complication rates in patient with SARS-CoV-2 infection and the typically CT-findings. I read the manuscript with interests, but some points coming up while reviewing the manuscript which have to address.

1 A critical point is the small number of patients. Were there any differences to complication rates in the group of ISS > 16 and with or without positive CT findings?

2 Respected to the SARS-CoV-2 virus variants, did you find differences? The study was finished before omicron was found. Do you think, that would be impacted the results?

3 What is with patients with other infections and traumata? Is it the same impact to pulmonary complication rates or outcome? What is with trauma specific pulmonary injury?

4 What is mean by the phrases “overall pulmonary complications”?

5 In the discussion section you described detailed the pathophysiology and therapeutic strategy of pulmonary embolism, whereas you found less or no relevant impact of embolism in your study population. That’s all standard of care and no relevant new findings. Please, state more to the obtain results.

6 The study aims clear formulated. Please respected them more in the discussion.

7 Line 114 to 120 The contents of the sentences are twice.

8 A limitation section is missing.

9 A flow chart of the study would be helpful for the readership.

10 Please provide an abbreviation list.

Author Response

Dear Sir or Madam

1) Exactly because of the small number of cases and no significant difference we decided not to include this in our study, because it makes little sense to compare 6 with 9 patients, here we would need another study with more patients.

2) Yes, included were the beta, gamma, and delta variants of SARS-CoV-2. It can be assumed that due to the overall milder courses, there would also be fewer complications in trauma patients. to verify this, one would have to perform a new study.

3)This question is valid, but our study did not have this objective. For this reason, we used the comparative group of trauma patients without infection to show what proportion of complications were trauma-related.

4) In "overall pulmonary complications", all of the 3 pulmonary complications (Pulmonary embolism, pneumonia, pleural effusion) are included

5) Thanks for the feedback, we have simplified and shortened the paragraph

6) We have written an additional section on the inpatient course of polytrauma patients with Sars-CoV-2 and compared this to cmoplication rates of polytrauma patients without covid

7) we corrected it to one sentence

8) Is written now

9) A flowchart is now included

10) In the instructions is written:

Abbreviations/Initialisms should be defined the first time they appear in each of three sections

We have followed this. However, if you still require a list, we can submit it in addition

Round 2

Reviewer 2 Report

Thanks for the revised version. Nothing to add in the present version. 

Author Response

Dear Sir or Madam

Thank you for your time and advices!

Kind regards

Fabrice Scheurer